# PROVABLE OPTIMAL TRANSPORT WITH TRANSFORMERS: THE ESSENCE OF DEPTH AND PROMPT ENGINEERING

## ABSTRACT

Can we establish provable guarantees for transformer performance? Providing such theoretical guarantees is a milestone in developing trustworthy generative AI. In this paper, we take a step toward addressing this question by focusing on optimal transport, a fundamental problem at the intersection of combinatorial and continuous optimization. Leveraging the computational power of attention layers, we prove that a transformer with fixed parameters can effectively solve the optimal transport problem (in Wasserstein-2 with entropic regularization) for an arbitrary number of points. Consequently, the transformer can sort lists of arbitrary size up to an approximation factor. Our results rely on an engineered prompt that enables the transformer to implement gradient descent with adaptive step sizes on the dual optimal transport. Combining the convergence analysis of gradient descent with Sinkhorn dynamics, we establish an explicit approximation bound for optimal transport with transformers, which improves with increasing depth. Our findings provide novel insights into the essence of prompt engineering and depth for transformers.

## 1 INTRODUCTION

Language models with *theoretical guarantees* are reliable and, therefore, more practical. Extensive experiments confirm the striking capabilities of transformers, such as "multi-task learning" (Radford et al., 2019), "in-context learning" (Brown, 2020), generalization Garg et al. (2022) to name but a few. But is it possible to **theoretically ensure** these capabilities and quantify their limits? Consider the simple example of sorting. We prompt the GPT-4 model to assess whether it can sort:

$$\text{prompt: sort}(2, 1, 4, 3) \rightarrow \text{output: } (1, 2, 3, 4)$$

While GPT-4 seems to be capable of sorting, querying to verify sorting is computationally infeasible for two reasons: (i) the elements in the list can be arbitrary numbers, and (ii) the list can be arbitrarily large. Thus, an infinite number of queries would be needed to verify that GPT-4 can sort. For theoretical verification of sorting, we need to study language models at a mechanistic level beyond black-box querying. Here, we investigate how to develop theoretical guarantees for the more general problem of *optimal transport*.

Optimal transport is a fundamental optimization problem at the intersection of combinatorial and continuous optimization. Sorting is a special case of optimal transport in one dimension (Brockett, 1991). While efficient sorting algorithms are discrete, Brockett (1991) raised the fundamental question of how to solve optimal transport with continuous dynamical systems. This question was motivated by the success of neural networks, which generate continuous state dynamics across their layers for feature extraction. Brockett (1991) proposes a continuous-state dynamical system over the orthogonal group that iteratively solves optimal transport and, hence, can sort and diagonalize matrices. Building upon this study, we investigate whether the feature dynamics in transformers are capable of performing optimal transport.

Beyond statistical parametric models, language models are powerful computational machines. Recent studies show that language models are capable of learning to implement algorithms, including gradient descent for least squares (Ahn et al., 2024; Von Oswald et al., 2023; Akyürek et al., 2022),

and the temporal difference method for reinforcement learning (Wang et al., 2024), and support vector mechanism (Tarzanagh et al., 2023). This computational perspective has provided valuable insights into language models at a mechanistic level, linking them to algorithms with provable guarantees. Using this approach, **we establish theoretical guarantees for solving optimal transport (in Wasserstein-2) with transformers**. Specifically, we prove that transformers can implement gradient descent (with adaptive step sizes) on the dual optimal transport objective regularized by entropy. In particular, each layer with two attention heads can simulate an iteration of gradient descent. Therefore, the induction of multiple attention heads can implement several iterations of gradient descent. This connection allows us to establish error bound for optimal transport with transformers that vanishes with *depth*: Given two sets of $n$ points in $\mathbb{R}^d$, a transformer can approximate the optimal transport map up to

$$O\left(\frac{n^{3/2}}{\text{depth}^{1/2}}\right) \text{-error for all integer } n. \tag{1}$$

Remarkably, the above bound holds for different choice of $n$ indicating that the transformer can solve multi-instances of optimal transport at the same time, an assertion for the capability of multi-task learning. Our results depend on the specific engineering of the prompt.

Recent studies demonstrate that interacting with language models is an art: proper prompting can significantly enhance their performance. The seminal work of Kojima et al. (2022) shows that adding phrases such as "let's think step by step" to the prompt encourages language models to produce more accurate reasoning. Prompting is becoming an essential skill in modern society, as prompt engineering positions are now being posted and well-paid by various companies. But what is the essence of prompt engineering? For the case study of optimal transport, we show that prompt engineering can significantly enhance the computational capabilities of transformers by providing the necessary memory and statistics.

## 2 BACKGROUND

### 2.1 OPTIMAL TRANSPORT

Consider two sets of points $x_1, \ldots, x_n \in \mathbb{R}^d$ and $y_1, \ldots, y_n \in \mathbb{R}^d$, and define $C \in \mathbb{R}^{n \times n}$ such that $C_{ij} = \|x_i - y_j\|_2^2$. Finding the optimal transport map (in Wasserstein-2 metric) between these two sets of points casts to the optimization of a linear function over the set of permutation matrices (Cuturi, 2013):

$$P_* := \arg\min_{P \in \mathbb{R}^{n \times n}} \text{Tr}(PC), \quad \text{subject to } P \text{ being a } permutation \text{ matrix}, \tag{2}$$

where $\text{Tr}(M)$ denotes the trace of matrix $M$. Sorting lists is an example of optimal transport. Specifically, if $C_{ij} = (x_i - y_j)^2$, where $y_i = i$ for $i = 1, \ldots, n$ and $x_1, \ldots, x_n \in \mathbb{R}$, then the linear transformation of $[x_1, \ldots, x_n]$ with $P_*$ sorts $x_1, \ldots, x_n$ (see Remark 2.28 in (Peyré et al., 2019) for more details). Yet, the optimal transport problem is more general than sorting.

Optimal transport lies at the intersection of discrete and continuous optimization. There are various combinatorial algorithms for sorting and optimal transport, but our primary focus here is on continuous optimization methods, which allow us to understand the mechanism of transformers. In particular, we review two fundamental methods: (i) constrained continuous optimization and (ii) Sinkhorn regularization.

While optimal transport involves optimization over the combinatorial set of permutation matrices, it can be relaxed to optimization over a continuous set. The state-of-the-art method is based on linear programming, specifically solving the following continuous convex optimization problem:

$$\widehat{P} = \arg\min_{P \in \mathbb{R}^{n \times n}} \text{tr}(PC), \quad \text{subject to } P \text{ being a } doubly \text{ } stochastic \text{ matrix}.$$

Comparing the above problem with the original problem in (2), we notice that the constraint requiring $P$ to be a permutation matrix has been relaxed to allowing $P$ to be a doubly stochastic matrix. Recall the solutions to linear programs lie among the extreme points of the constraint set. Since the extreme points of doubly stochastic matrices are permutation matrices (Conte et al., 1991), the above linear program has the same solution as (2), i.e., $P_* = \widehat{P}$.

The above linear program requires the optimization of $O(n^2)$ variables. Due to the quadratic growth with $n$, solving the linear program becomes computationally challenging for large $n$. Cuturi (2013) proposes a computationally efficient alternative based on regularization with entropy:

$$P_\gamma^* := \arg \min_{P \in \mathbb{R}^{n \times n}} \text{Tr}(PC) + \lambda \sum_{ij} P_{ij} \log(P_{ij}), \text{ subject to } P \text{ is a } \textit{doubly stochastic} \text{ matrix} \quad (3)$$

The Lagrangian dual of the above program reduces to the optimization of $O(n)$ variables which is considerably fewer than $O(n^2)$ variables for the original linear program. Introducing the dual parameters $v \in \mathbb{R}^n$ and $u \in \mathbb{R}^n$, the Lagrangian function is defined as follows:

$$L(u, v, C) = \text{Tr}(PC) + \lambda \sum_{ij} P_{ij} \log(P_{ij}) - u^\top \left( P1_n - \frac{1}{n}1_n \right) - v^\top \left( P^\top 1_n - \frac{1}{n}1_n \right).$$

It is easy to check that the minimizer of $L$ with respect to $P$ is $P_{ij} = e^{\frac{-C_{ij} + v_j + u_i}{\lambda} - 1}$. This structure inspired the use of Sinkhorn's fixed point iteration to find the solution of the dual problem. In particular, (Sinkhorn, 1967) proves that there exists a unique doubly stochastic matrix of the form $[P_\lambda^*]_{ij} = e^{\frac{-C_{ij} + v_i^* + u_j^*}{\lambda} - 1}$ that is the solution of a simple fixed point iteration where $u^*, v^*$ are unique up to scaling factors. Leveraging this fundamental theorem, Cuturi (2013) proposes a fixed-point iteration that efficiently solves the dual problem. We will later elaborate on the fixed-point iteration and its convergence.

Apart from Sinkhorn's fixed point iteration, there are many different methods to solve Lagrangian dual problem such as first-order optimization methods. Recall the minimizer $P_{ij} = e^{\frac{-C_{ij} + u_i + v_j}{\lambda} - 1}$. Plugging the minimizer into $L$ reduces the problem to the following optimization

$$\arg \min_{v, u \in \mathbb{R}^n} \left( L(u, v) := \lambda \left( \sum_{ij} e^{(-C_{ij} + u_i + v_j)/\lambda - 1} \right) - \frac{1}{n} \sum v_i - \frac{1}{n} \sum_i u_i \right).$$

It is easy to check that $L$ is convex in $u$ and $v$ as its Hessian is diagonally dominant, hence positive semi-definite. Thus, standard first-order optimization can optimize $L$. In particular, one can use gradient descent (with adaptive stepsizes), such as the following recurrence

$$\begin{cases} u^{(\ell+1)} = u^{(\ell)} - D_\ell \nabla_u L(u^{(\ell)}, v^{(\ell)}) \\ v^{(\ell+1)} = v^{(\ell)} - D_\ell' \nabla_v L(u^{(\ell)}, v^{(\ell)}) \end{cases}, \quad (4)$$

where $\nabla_u L$ denotes the gradient of $L$ with respect to $u$ and $D_\ell, D_\ell' \in \mathbb{R}^{n \times n}$ are are diagonal matrices with positive diagonal elements. We will prove that self-attention layers can implement the above recurrence.

## 2.2 SELF-ATTENTION LAYERS

Attention layers are fundamental building blocks of neural networks, developed over decades of research. Hochreiter (1997) pioneered this development by proposing an attention mechanism for Recurrent Neural Networks (RNNs) inspired by human cognition. Graves (2014) employs the attention mechanism to develop a memory system for a parametric version of the Turing machine. Bahdanau (2014) adapts this attention mechanism in neural Turing machines to design a powerful model for machine translation. While attention was originally introduced for recurrent models, Vaswani (2017) introduced non-recurrent attention layers, combined with residual connections (He et al., 2016), thereby significantly enhancing the training of attention weights.

Attention layers rely on on a convex combination. Let $Z \in \mathbb{R}^{m \times d}$. An attention layer is a function denoted by $\text{atten}_w : \mathbb{R}^{m \times d} \to \mathbb{R}^{m \times d}$ with parameters $w := \left[ w_k, w_v, w_q \in \mathbb{R}^{d \times d} \right]$ defined as

$$\text{atten}_{[w]}(Z) = A_{m \times m} Z w_v, \quad A_{ij} = \frac{e^{\langle w_k z_i, w_q z_j \rangle}}{\sum_{j=1}^m e^{\langle w_k z_i, w_q z_j \rangle}}, \quad (5)$$

where $z_i$ and $z_j$ are rows of $Z$. The convex combination of data points imposes a local dependency that can simulate the focusing mechanism in neural networks.

Tay et al. (2020) investigates whether attention layers are capable of sorting. Since self-attention layers cannot directly implement Sinkhorn's fixed-point iteration, Tay et al. (2020); Sander et al. (2022) propose a novel attention mechanism called "Sinkhorn attention". However, we demonstrate that standard attention layers can implement gradient descent with adaptive step sizes on $L$.

## 3 PROMPTING AND MODEL

**The Engineered Prompt.** We propose a particular prompt structure to encode optimal transport in transformers:

$$
Z_n = \begin{bmatrix} x_1 & y_1 & \|x_1\|^2 & \|y_1\|^2 & 1 & 1 & 1 & 1 & 0 & 0 & 0 \\ x_2 & y_2 & \|x_2\|^2 & \|y_2\|^2 & 1 & 1 & 1 & 1 & 0 & 0 & 0 \\ \vdots & \vdots & \vdots & \vdots & \vdots & \vdots & \vdots & \vdots & \vdots & \vdots & \vdots \\ x_n & y_n & \|x_n\|^2 & \|y_n\|^2 & 1 & 1 & 1 & 1 & 0 & 0 & 0 \\ 0 & 0 & 0 & 0 & 0 & 0 & 0 & -1/n & 0 & 0 & 0 \end{bmatrix} \in \mathbb{R}^{(n+1)\times(2d+9)}. \tag{6}
$$

The highlighted elements in blue are the original prompts, which are sufficient for optimal transport. The elements highlighted in red are carefully engineered. We will prove that this particular prompt engineering allows attention layers to iteratively solve optimal transport.

**Transformer.** We consider a specific transformer architecture composed of multiple attention and feedforward layers, all connected via residual connections. Let $Z^{(\ell)}$ denote the intermediate representation of the input $Z$ at layer $\ell$ which obeys the following recurrence

$$
Z_n^{(0)} = Z_n,
$$

$$
Z_n^{(\ell+1/2)} = Z_n^{(\ell)} + \sum_{j=1}^{2} \text{atten}_{w^{(\ell,j)}}(Z_n^{(\ell)}) B_j^{(\ell)}, \tag{7}
$$

$$
Z_n^{(\ell+1)} = Z_n^{(\ell+1/2)} + (Z_n^{(\ell+1/2)} w_f^{(\ell)})_+,
$$

where $w_f^{(\ell)} \in \mathbb{R}^{d'\times d'}$ are the weight matrices for the feedforward layers, $B_j^{(\ell)} \in \mathbb{R}^{d'\times d'}$ are the mixing weights for the attention heads, and $(a)_+ = \max(0, a)$ represents the ReLU activation function used in the feedforward layers. The model includes two attention heads and employs the standard softmax attention mechanism commonly used in practice. This makes our model more closely aligned with practical transformer architectures, in contrast to previous theoretical studies focusing on linear attention layers (Ahn et al., 2024; Wang et al., 2024).

## 4 TRANSFORMERS AS ITERATIVE ALGORITHMS

### 4.1 ADAPTIVE GRADIENT DESCENT WITH TRANSFORMER

We prove that transformers can implement iterations of gradient descent. The proof relies on the expressive power of attention layers combined with the engineered prompt, which provides the required memory to store iterates of gradient descent.

**Theorem 1.** *There exists a configuration of parameters such that*

$$
\begin{cases} [Z_n^{(\ell)}]_{(1:n),(2d+7)} = u^{(\ell)} - D_\ell \nabla_u L(u^{(\ell)}, v^{(\ell)}) \\ [Z_n^{(\ell)}]_{(1:n),(2d+8)} = v^{(\ell)} - D'_\ell \nabla_v L(u^{(\ell)}, v^{(\ell)}) \end{cases},
$$

*holds for all integer values of* $n$, *where* $u^{(\ell)}$ *and* $v^{(\ell)}$ *are gradient descent in* (4) *iterations starting from* $u_0 = v_0 = 0$ *with the following adaptive stepsizes*

$$
[D_\ell]_{ii} = \frac{\gamma_\ell}{\sum_j e^{(-C_{ij}+u_i^{(\ell)}+v_j^{(\ell)})/\lambda-1} + 1}, \quad [D'_\ell]_{jj} = \frac{\gamma_\ell}{\sum_i e^{(-C_{ij}+u_i^{(\ell)}+v_j^{(\ell)})/\lambda-1} + 1}.
$$

Remarkably, the above result holds for **an arbitrary** $n$ since a transformer can accept inputs of varying sizes. Indeed, a single transformers with a constant parameters can implement GD with adaptive stepsizes for optimal transport of arbitrary input size $n$. We will elaborate on this important property by establishing convergence rate to $P_\lambda^*$, thereby proving that a single transformer is capable of solving optimal transport for all $n$.

Notably, the above result supports the *"iterative inference hypothesis"* (Jastrzębski et al., 2017), that links the mechanism of deep neural networks to widely used optimization methods. This hypothesis

posits that residual connections enable deep networks to implicitly implement gradient descent across layers to tackle complex tasks. It is based on striking observations on the underlying mechanisms of Convolutional Neural Networks (CNNs) (Alain, 2016). Previous studies have theoretically proven this hypothesis for solving least-squares problems (Ahn et al., 2024; Von Oswald et al., 2023; Akyürek et al., 2022) using transformers. Building on these studies, we demonstrate that transformers can implement gradient descent for a different objective function to solve optimal transport, advancing our understanding of the iterative inference mechanism in deep networks

Prompt engineering is essential for the proof. Expanding the input size by adding columns and rows creates an extended data representation matrix across the layers. Attention layers can utilize a part of this expanded matrix as memory to store the iterates of gradient descent. Furthermore, the input dependent part of the prompt supplies the necessary statistics for the attention layers to implement gradient descent. To elaborate, we will present the proof and explain the essence of attention layers and prompt engineering.

### 4.2 PROOF OF THEOREM 1

The proof leverages the computational power of attention layers. We demonstrate that two attention heads can jointly implement a single step of gradient descent (with adaptive step sizes) on $L(u, v)$. By induction, multiple attention heads can implement several iterations of gradient descent with adaptive step sizes. The proof is constructive, explicitly determining the choice of parameters.

**Parameters Choice.** Define $Q^{(\ell,j)} = w_k^{(\ell,j)}(w_q^{(\ell,j)})^\top$. Let $d' = 2d + 9$ and $e_i \in \mathbb{R}^{d'}$ denote the $i$-th standard basis vector $[e_i]_j = \begin{cases} 1 & i = j \\ 0 & \text{otherwise} \end{cases}$. We choose parameters such that

$$\lambda Q^{(\ell,1)} = [\mathbf{0}_{d' \times d} \quad 2e_1, \dots, 2e_d \quad \mathbf{0}_{d'} \quad -e_{2d+3} \quad -e_{2d+1} \quad e_{2d+7} \quad \mathbf{0}_{d'} \quad -\lambda e_{2d+5} \quad \mathbf{0}_{d'} \quad \mathbf{0}_{d'} \quad e_{2d+5} \quad \mathbf{0}_{d'}] \in \mathbb{R}^{d' \times d'}$$

$$\lambda Q^{(\ell,2)} = [2e_{d+1}, \dots, 2e_{2d} \quad \mathbf{0}_{d' \times d} \quad -e_{2d+3} \quad \mathbf{0}_{d'} \quad -e_{2d+2} \quad e_{2d+8} \quad -\lambda e_{2d+5} \quad \mathbf{0}_{d'} \quad e_{2d+5} \quad \mathbf{0}_{d'} \quad \mathbf{0}_{d'}] \in \mathbb{R}^{d' \times d'}$$

$$[w_v^{(\ell,1)}]_{ij} = \begin{cases} 1 & i = 2d + 6 \text{ and } j = 2d + 7 \\ 0 & \text{otherwise} \end{cases}, \quad w_f^{(\ell)} = \mathbf{0}_{d' \times d'}$$

$$[w_v^{(\ell,2)}]_{ij} = \begin{cases} 1 & i = 2d + 6 \text{ and } j = 2d + 8 \\ 0 & \text{otherwise} \end{cases}, \quad B_j^{(\ell)} = \gamma_\ell I_{d' \times d'}.$$

(8)

Notably, there are many choices for $w_k^{(\ell,j)}$ and $w_q^{(\ell,j)}$ that ensure the above equations hold.

**Notations.** Consider the matrix $M \in \mathbb{R}^{n \times n}$ defined as $M_{ij} = e^{\frac{-C_{ij} + u_i + v_j}{\lambda} - 1}$. By definition,

$$L(u, v) = \lambda \sum_{ij} \underbrace{\exp\left(\frac{-C_{ij} + u_i + v_j}{\lambda} - 1\right)}_{=M_{ij}} - \frac{1}{n}\sum_i u_i - \frac{1}{n}\sum_j v_j,$$

holds for $C_{ij} = \|x_i - y_j\|^2 = x_i^2 + y_j^2 - 2\langle x_i, y_j \rangle$.

**A Proof Based on Induction.** Assuming that the statement holds for $\ell$, we prove that it holds for $\ell + 1$. It is easy to check that induction base holds. The choice of $w_f^\ell$ and $w_v^{(\ell,j)}$ ensure that only the $2d + 7$-th and $2d + 8$-th columns of $Z_n^{(\ell)}$ change with $\ell$. Induction hypothesis concludes

$$Z_n^{(\ell)} = \begin{bmatrix} x_1 & y_1 & \|x_1\|^2 & \|y_1\|^2 & 1 & 1 & 1 & 1 & u_1^{(\ell)} & v_1^{(\ell)} & 0 \\ x_2 & y_2 & \|x_2\|^2 & \|y_2\|^2 & 1 & 1 & 1 & 1 & u_2^{(\ell)} & v_2^{(\ell)} & 0 \\ \vdots & \vdots & \vdots & \vdots & \vdots & \vdots & \vdots & \vdots & \vdots & \vdots & \vdots \\ x_n & y_n & \|x_n\|^2 & \|y_n\|^2 & 1 & 1 & 1 & 1 & u_n^{(\ell)} & v_n^{(\ell)} & 0 \\ 0 & 0 & 0 & 0 & 0 & 0 & 0 & -1/n & ? & ? & 0 \end{bmatrix} \in \mathbb{R}^{(n+1) \times (2d+9)},$$

where elements highlighted in teal indicates the equality that follows from the induction assumption. Assuming the inequality above holds, we proceed to prove that

$$
Z_n^{(\ell)} = \begin{bmatrix}
x_1 & y_1 & \|x_1\|^2 & \|y_1\|^2 & 1 & 1 & 1 & 1 & u_1^{(\ell+1)} & v_1^{(\ell+1)} & 0 \\
x_2 & y_2 & \|x_2\|^2 & \|y_2\|^2 & 1 & 1 & 1 & 1 & u_2^{(\ell+1)} & v_2^{(\ell+1)} & 0 \\
\vdots & \vdots & \vdots & \vdots & \vdots & \vdots & \vdots & \vdots & \vdots & \vdots & \vdots \\
x_n & y_n & \|x_n\|^2 & \|y_n\|^2 & 1 & 1 & 1 & 1 & u_n^{(\ell+1)} & v_n^{(\ell+1)} & 0 \\
0 & 0 & 0 & 0 & 0 & 0 & 0 & -1/n & ? & ? & 0
\end{bmatrix} \in \mathbb{R}^{(n+1)\times(2d+9)}.
$$

Indeed, the extended prompt offers sufficient memory to store the vectors $u^{(\ell)}$ and $v^{(\ell)}$ obtained through gradient descent on $L(u, v)$.

**Constructing Gradients With Attention Heads.** We begin by computing the output of the first attention head in layer $\ell + 1$, step by step. Through straightforward algebra, we obtain the following:

$$
Z_n^{(\ell)} Q^{(\ell,1)} = \begin{bmatrix}
0 & 2x/\lambda & \mathbf{0}_n & -\mathbf{1}_n/\lambda & -\|x\|^2/\lambda & u^{(\ell)}/\lambda & 0 & -\mathbf{1}_n & \mathbf{0}_n & \mathbf{0}_n & \mathbf{1}_n/\lambda & 0 \\
0 & 0 & 0 & 0 & 0 & 0 & 0 & 0 & 0 & 0 & 0
\end{bmatrix} \in \mathbb{R}^{(n+1)\times d'}
$$

where $x = [x_1, \ldots, x_n] \in \mathbb{R}^{n\times d}$ and $x^2 = [\|x_1\|^2, \ldots, \|x_n\|^2] \in \mathbb{R}^n$. This equation results in the following:

$$
Z_n^{(\ell)} Q^{(\ell,1)} (Z_n^{(\ell)})^\top = \begin{bmatrix}
\frac{1}{\lambda}\left( -x^2 \mathbf{1}_n^\top + 2xy^\top - \mathbf{1}_n(y^2)^\top + u^{(\ell)}\mathbf{1}_n^\top + \mathbf{1}_n(v^{(\ell)})^\top \right) - \mathbf{1}_n\mathbf{1}_n^\top & \mathbf{0}_n \\
\mathbf{0}_n^\top & 0
\end{bmatrix}
$$

$$
= \begin{bmatrix}
\log(M_{n\times n}) & \mathbf{0}_n \\
\mathbf{0}_n^\top & 0
\end{bmatrix}
$$

where $\log(M)$ for an input matrix $M$ is defined as a matrix with $[\log(M)]_{ij} = \log(M_{ij})$. Similarly, we define the matrix $\exp(M)$ such that $[\exp(M)]_{ij} = e^{M_{ij}}$. Thus,

$$
\exp(Z_n^{(\ell)} Q^{(\ell,1)} (Z_n^{(\ell)})^\top) = \begin{bmatrix}
M_{n\times n} & \mathbf{1}_n \\
\mathbf{1}_n^\top & 1
\end{bmatrix} \tag{9}
$$

Furthermore, the choice of parameters $w_v^{(\ell,1)}$ obtains

$$
Z_n^{(\ell)} w_v^{(\ell,1)} = -\gamma \begin{bmatrix}
\mathbf{0}_n & \cdots & \mathbf{0}_n & \mathbf{1}_n & \mathbf{0}_n & \mathbf{0}_n \\
0 & \cdots & 0 & -1/n & 0 & 0
\end{bmatrix}
$$

Stitching all equations together yields

$$
\mathrm{atten}_{w^{(\ell,1)}}(Z_n^{(\ell)}) B_1^{(\ell)} = \begin{bmatrix}
\mathbf{0}_n & \cdots & \mathbf{0}_n & -D_\ell(M\mathbf{1}_n - \frac{1}{n}\mathbf{1}_n) & \mathbf{0}_n & \mathbf{0}_n \\
0 & \vdots & 0 & n-1/n & 0 & 0
\end{bmatrix}
$$

Similarly, we can show that

$$
\mathrm{atten}_{w^{(\ell,2)}}(Z_n^{(\ell)}) B_2^{(\ell)} = \begin{bmatrix}
\mathbf{0}_n & \cdots & \mathbf{0}_n & \mathbf{0}_n & -D_\ell'(M^\top\mathbf{1}_n - \frac{1}{n}\mathbf{1}_n) & \mathbf{0}_n \\
0 & \vdots & 0 & 0 & n-1/n & 0
\end{bmatrix}
$$

Replacing the last two equations into (7) concludes the induction proof.

## 5  PROVABLE OPTIMAL TRANSPORT WITH TRANSFORMERS

By linking the intermediate data representation to a well-established algorithm, we gain access to powerful theoretical tools to prove that transformers can solve optimal transport. In particular, we utilize convergence analysis for gradient descent as well as Sinkhorn's recurrence.

**Linking Attention Patterns to the Optimal Transport Map.** Recall the optimal transport map $P_\lambda^*$ defined in (3). We will prove that this matrix can be estimated by the attention matrices across the layers. Consider a block of *attention patterns* denoted by $A^{(\ell)} \in \mathbb{R}^{n \times n}$, defined as

$$A_{ij}^{(\ell)} = \frac{e^{\langle w_k^{(\ell,1)} z_i^{(\ell)}, w_q^{(\ell,1)} z_j^{(\ell)}\rangle}}{\sum_{j=1}^n e^{\langle w_k^{(\ell,1)} z_i^{(\ell)}, w_q^{(\ell,1)} z_j^{(\ell)}\rangle}}, \tag{10}$$

where $z_i^{(\ell)}$ is the $i$-th row of $Z^{(\ell)}$. We establish the convergence of $A^{(\ell)}$ to $P_\lambda^*$ in an appropriate metric.

**Convergence Proof.** As discussed, the optimal transport matrix $P_\lambda^*$ has the following form (Cuturi, 2013):

$$P_\lambda^* = \mathrm{diag}(w^*) Q \mathrm{diag}(q^*), \quad w^*, q^* \in \mathbb{R}_+^n, \quad Q \in \mathbb{R}_+^{n \times n}, \quad Q_{ij} = e^{-\frac{C_{ij}}{\lambda} - 1}.$$

It is easy to verify that replacing $w^*$ and $q^*$ with $cw^*$ and $q^*/c$ leads to the same matrix $P_\lambda^*$ for all $c \in \mathbb{R}_+$. Franklin and Lorenz (1989) introduce a metric that accounts for this particular scaling invariance. Consider the following metric

$$\mu(w, w') = \log\left(\max_{ij} \frac{w_i w_j'}{w_j w_i'}\right). \tag{11}$$

Remarkably, $\mu$ is a metric that satisfies the triangle inequality (Franklin and Lorenz, 1989). However, $\mu$ is not a norm, as $\mu(w, w') = 0$ only implies that there exists a constant $c$ such that $w = cw'$. The next theorem establishes an explicit convergence rate for the attention matrices $A^{(\ell)}$ to the solution of optimal transport in $\mu$.

**Theorem 2.** *There exists a choice of parameters and an integer $k \le \ell$ such that $A^{(k)}$ can be expressed as:*

$$A^{(k)} = diag(w^{(k)}) Q diag(q^{(k)}),$$

*where $w^{(k)}, q^{(k)} \in \mathbb{R}_+^n$ obey:*

$$\max\left\{\mu(q^{(k)}, q^*), \mu(w^{(k)}, w^*)\right\} \le \frac{36 n^{3/2} e^{r/\lambda} \sqrt{r}}{\sqrt{\ell}(1 - \eta)},$$

*for $(1/4) r^2 = \|w^{(1)} - w^*\|_2^2 + \|q^{(1)} - q^*\|_2^2, \eta = (\phi(Q)^{1/2} - 1)/(\phi(Q)^{1/2} + 1), \phi(Q) = \max_{ijkl} Q_{ik} Q_{jl}/(Q_{jk} Q_{il})$, as long as $\ell \ge 64 n^3 \exp(3r/\lambda) r$.*

The above theorem theoretically verifies that transformers can solve optimal transport for an arbitrary number of points $n$, with provable worst-case approximation guarantees. According to the theorem, the attention patterns converge to the optimal transport matrix $P_\lambda^*$ at a rate of $O\left(\frac{1}{\mathrm{depth}^{1/2}}\right)$, implying that the transformer performance improves with increasing depth.

An application of the last theorem is the provable sorting capability of transformers, achieved up to certain approximation factors. As discussed, sorting is a specific case of optimal transport for $d = 1$ with $y_1 \le \cdots \le y_n$. By solving the optimal transport problem, transformers can effectively sort lists. Since convergence is guaranteed for regularized transportation, we expect the transformer to sort with an error that diminishes as $\lambda \to 0$. We will experimentally assess the sorting accuracy with the transformer in experiments.

The statement of the last theorem proves that it is possible to avoid the rank collapse of attention layers, which indicate that attention patterns converge to a certain low-rank matrix as depth increases (Geshkovski et al., 2024; Wu et al., 2024; Dong et al., 2021). Dong et al. (2021) shows that attention layers without residual connection suffer from this collapse. Geshkovski et al. (2024) proves even with residual connections attention with symmetric weights may suffer from the rank collapse. This rank collapse significantly reduces the expressivity of attention layers and poses challenges for training (Daneshmand et al., 2020). Consequently, a line of research investigates effective techniques to avoid the rank collapse issue (Meterez et al., 2023; Daneshmand et al., 2021; Joudaki et al., 2023). We argue that the last theorem demonstrates the possibility of avoiding the rank collapse of attention patterns with a specific prompt engineering and particular parameter choices. Recall that $P_\lambda^*$ serves as an approximation of the optimal transport matrix $P^*$, which is a full-rank permutation matrix. Consequently, we anticipate that the attention patterns will retain a high rank throughout the layers, as will be demonstrated in the experiments.

## 6 EXPERIMENTS

While our contribution is primarily theoretical, we connect the theoretical findings to practical observations through experiments. First, we experimentally validate our findings in Theorems 1 and 2, proving that transformers are capable of optimal transport. Second, we show that transformers can learn from data to solve optimal transport, combining the theoretical expressivity with data-driven learning used in practice. Finally, we show the significance of prompt engineering in practice.

**Data specification.** We consider optimal transport with $d = 1$, varying $n$, which is the focus of our study. In particular, $x_1, \ldots, x_n$ are a random permutation of $[1/n, 2/n, \ldots, n/n]$, and $y_i = i/n$ in our experiments. We use a regularization constant of $\lambda = 0.005$ in the related experiments.

**Training protocol.** In Sections 6 and 6, we use Adam (**?**) with a step size of $0.001$ and $10^4$ iterations for training. We reparameterize $Q_\ell := w_k^{(\ell)}(w_q^{(\ell)})^\top$ and optimize $Q_\ell$. Parameters are initialized randomly from a Gaussian distribution with variance $1/(2d+9)$. Notably, we set $w_f^{(\ell)} = 0$, $B_j^{(\ell)} = (1/20)I_{d'}$, and only optimize $w^{(\ell,j)} := [w_k^{(\ell,j)}, w_q^{(\ell,j)}, w_v^{(\ell,j)}]$, as this does not limit the optimal transport capability (see (8) ). Training is conducted on a single T4 GPU.

**Validations.** Since the proof of Theorem 1 is constructive, it provides an explicit choice of parameters for transformers. Specifically, we set $\gamma_k = 0.01$ and use the weight matrices defined in (8). We examine the sorting accuracy of the transformer. As previously discussed, sorting is equivalent to optimal transport for $d = 1$. Given the attention pattern $A^{(2000)}$, we compute the linear transformation of the input $x = [x_1, \ldots, x_n]$ using $A^{(2000)}$ to generate an estimate for the sorted $x$. An example of sorting is:

$$\text{Input} : [0.5, 0.75, 0.25, 0.0] \quad \rightarrow \quad \text{Output} : [0.018, 0.24, 0.50, 0.73]$$

We observe that generated output approximates the sorted list $[0.0, 0.25, 0.5, 0.75]$. Interestingly, the same transformers can also approximately sorts larger lists:

$$[0.375, 0.5, 0.125, 0.875, 0.75, 0.25, 0.0, 0.625] \quad \rightarrow \quad [0.02, 0.12, 0.25, 0.37, 0.5, 0.62, 0.75, 0.84]$$

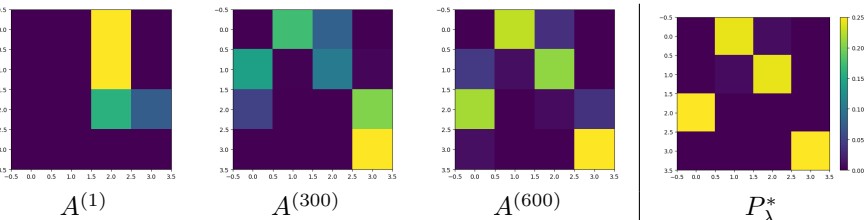

$$A^{(1)} \qquad A^{(300)} \qquad A^{(600)} \qquad P_\lambda^*$$

Figure 1: **Convergence of Attention Patterns.** The plotted matrices are $A^{(1)}$, $A^{(300)}$ and $A^{(600)}$ defined in (10). We observe these matrices converge to $P_\lambda^*$ (the rightmost plot). Thm 2 proves this convergence.

Figure 1 illustrates the convergence of the attention matrices $A^{(\ell)}$ to $P_\gamma^*$, as established in Theorem 2. Notably, we observe that the attention patterns maintain high rank across in contrast to the observations on the rank collapse phenomenon; please refer to remarks in Section 5.

Figure 2 further illustrates that a single network can solve optimal transport on different sample sizes. In particular, this figure demonstrates that the transformer, with the parameter choices specified in (8), can find $P_\gamma^*$ for both $n = 4$ and $n = 8$ simultaneously without any changes to the parameters.

**Training.** So far, we proved and experimentally validate that a transformers is capable of optimal transportation. Now, we experimentally investigate whether a transformer can learn from data to solve optimal transport. Recall the hidden representation of transformer at layer $\ell$ denoted by $Z_\ell$. We optimize parameters $w^{(\ell,j)}$ to solve the following minimization problem

$$\min_{w^{(1,j)}, \ldots, w^{(20,j)}} \mathbb{E}\left[\|[Z_{20}]_{1:n, 2d+9} - \text{sorted}(x)\|^2\right] \qquad \text{(training loss)}$$

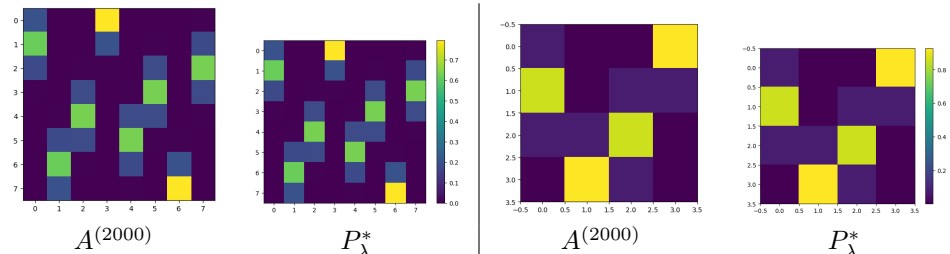

Figure 2: **Optimal Transport of Different Sizes.** left: $n = 8$, right: $n = 4$. The transformer weights remain exactly the same.

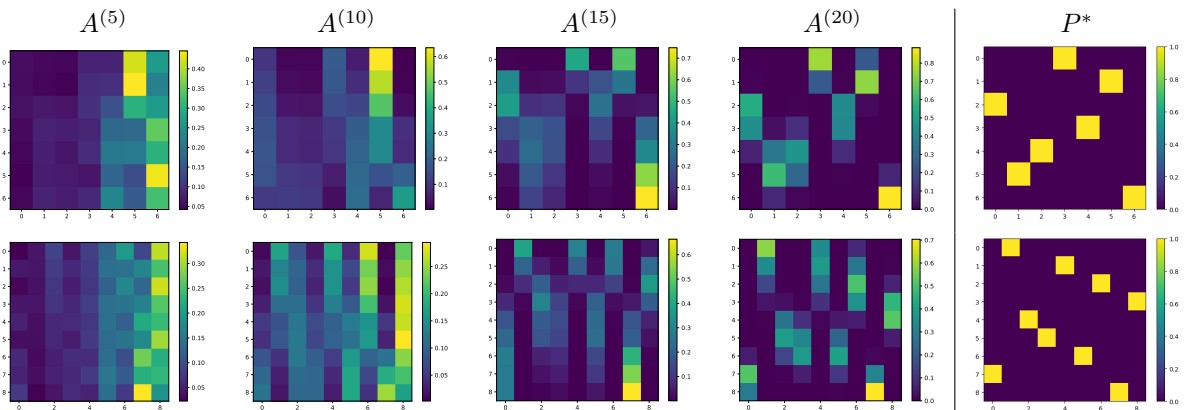

Figure 3: **Optimal Transport After Training.** *Rows:* $n = 7$ and $n = 9$; *Columns:* attention patterns $A^{(\ell)}$ defined in (10) for $\ell = 5, 10, 15, 20$. The last columns shows the optimal transport map, i.e., $P^*$ in (2) ; *Training:* optimizing the training loss on random data with $n = 7$ (see Section 6 for more details).

where sorted$(x) \in \mathbb{R}^n$ contains sorted $x_i$s and the expectation is taken over random data (details in **Data specification**). In order to approximate the expectation, we draw 500 samples uniformly at random. Figure 3 compares the attention patterns —denoted by $A^{(\ell)}$ defined in (10)— across the layers where we observe that these patterns are converging to the optimal solution. This observation validates that transformers iteratively solve optimal transport across their layers (similar to gradient descent on $L$). While the the transformer is trained for $n = 7$, we observe a good approximation for $n = 9$.

**Prompt engineering.** We experimentally evaluate the impact of the engineered prompt (6) on solving optimal transport for $d = 1$. Specifically, we reduce the number of columns in the prompt by removing additional ones as

$$Z' = \begin{bmatrix} x_1 & y_1 & 0 \\ \vdots & \vdots & \vdots \\ x_n & y_n & 0 \end{bmatrix} \in \mathbb{R}^{n \times 3}. \tag{12}$$

The last column is designated for the output. Let $Z'_{20}$ represent the output of a transformer with 20 layers. We optimize the weights so that the last column of $Z'_{20}$ predicts the sorted values of $x_1, \ldots, x_n$:

$$\min_{w^{(1,j)}, \ldots, w^{(20,j)}} \mathbb{E}\left[\|[Z'_{20}]_{:,3} - \text{sorted}(x)\|^2\right] \tag{13}$$

where details on training data and process are presented in **Data specification** and **Training protocol**, respectively.

Figure 4 shows the clear impact of prompt engineering on the performance, where the above prompt (without engineering) leads to a significantly worse approximate of optimal transport matrix.

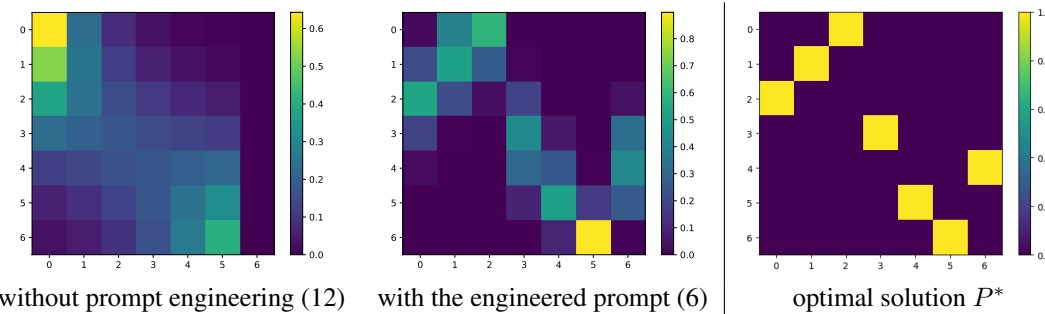

| without prompt engineering (12) | with the engineered prompt (6) | optimal solution $P^*$ |

Figure 4: **The Significance of Prompt Engineering.** *Left:* attention pattern in the last layer denoted by $A^{(20)}$ after optimizing the training loss (13) for inputs without prompt engineering; *Middle:* attention pattern in the last layer ($A^{(20)}$) after training with the engineered prompt in (6); *Right:* optimal transport map associated with the input. $x_1, \ldots, x_n$ and $y_1, \ldots, y_n$ in prompts are fixed.

## 7 DISCUSSIONS

We proved that transformers with fixed parameters can solve multiple instances of optimal transport on different number of points, with an explicit accuracy bound. Our analysis shows that transformers can implement gradient descent on a specific objective function using a specially engineered prompt. The engineered prompt provides additional memory to implement gradient descent. These findings open several avenues for future research, including: including: (i) depth-efficient guarantees, (ii) the analysis of training dynamics, and (iii) studying prompt engineering beyond optimal transport.

**(i) Depth Efficient Guarantees.** According to Theorem 2, a transformer with $O(\epsilon^{-2})$-depth can obtain an $O(\epsilon)$-accurate solution. This is due to the established convergence rate for gradient descent with adaptive stepsizes. However, $O(\log(1/\epsilon))$ Sinkhorn iterations suffice for achieving $\epsilon$-accuracy. While there is a considerable gap between the established convergence analysis for gradient descent and the convergence rate of Sinkhorn's iteration, our result is sufficient to prove deep transformers can provably solve optimal transport. We call for bridging this gap through a tighter convergence analysis.

**(ii) Training for Optimal Transport.** We proved that transformers are able to provably solve optimal transportation and experimentally showed (in Section 6) that transformers can learn to solve optimal transport in $\mathbb{R}$ by training over random observations. Building upon this observation, we suggest a theoretical analysis of the training mechanism for optimal transport. To pursue this line of research, one can check whether parameters in (8) are local or global minimizers of the training loss. (Ahn et al., 2024; Wang et al., 2024) demonstrate that the properties of generative data distributions can be leveraged to analyze the stationary points of training dynamics. We believe that this technique can be used to analyze the landscape of training loss for optimal transport.

**(iii) Prompt Engineering Beyond Optimal Transport.** Prompt engineering is essential for demonstrating that transformers are capable of solving optimal transport. In Section 6, we experimentally show that prompt engineering is also important in practice. Despite its widespread use, the underlying mechanisms of prompt engineering remain understudied. Studying prompt engineering for optimal transport is a step towards the broader goal of *understanding the role of prompt engineering in general*. We conjecture that prompt engineering enhances the computational power of transformers, enabling them to simulate a wider class of algorithms.

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
