## A  PROOF OF LEMMA 1

**(i) Convergence Analysis of Gradient Descent**   Define the concatenated vector of iterates as

$$\theta_k = \begin{bmatrix} u^{(k)} \\ v^{(k)} \end{bmatrix},$$

and consider the following block diagonal matrix:

$$\Lambda_k = \begin{bmatrix} D_k & 0 \\ 0 & D'_k \end{bmatrix},$$

where $D_k$ and $D'_k$ are diagonal matrices at iteration $k$.

The recurrence relation of the iterates defined in (4) leads to the following inequality:

$$\|\theta_{k+1} - \theta^*\|^2_{\Lambda_k^{-1}} = \|\theta_k - \theta^*\|^2_{\Lambda_k^{-1}} - 2\langle \theta_k - \theta^*, \nabla L(\theta_k)\rangle + \|\nabla L(\theta_k)\|^2_{\Lambda_k}, \tag{14}$$

where the weighted norm is defined as $\|v\|^2_A = v^\top A v$.

**Bounding the Matrices:** Define the ball $B(r) = \{\theta \in \mathbb{R}^n \mid \|\theta\| \leq r\}$. Assume that $\theta_k, \theta^* \in B(r)$. It can be verified that

$$\frac{\gamma_k}{n \exp(r/\lambda) + 1} I_n \preceq \Lambda_k \preceq \gamma_k I_n, \quad \text{and} \quad \nabla^2 L \preceq n \exp(r/\lambda) I_n, \tag{15}$$

where $\gamma_k = \frac{1}{n \exp(r/\lambda)}$.

**Smoothness of $L$:** Since $L$ is $n \exp(r/\lambda)$-smooth within $B(r)$, by Theorem 2.1.5 of Nesterov (2013), we have

$$\langle \nabla L(\theta), \theta - \theta^* \rangle \geq \frac{1}{n \exp(r/\lambda)} \| \nabla L(\theta) \|^2. \tag{16}$$

Substituting the above inequality into (14) yields

$$\| \theta_{k+1} - \theta^* \|^2_{\Lambda_k^{-1}} \leq \| \theta_k - \theta^* \|^2_{\Lambda_k^{-1}} - \left( \frac{2}{n \exp(r/\lambda)} - \gamma_k \right) \| \nabla L(\theta_k) \|^2. \tag{17}$$

**Monotonicity of $\Delta_k$.** Let $\Delta_k := \| \theta_k - \theta^* \|^2_{\Lambda_k^{-1}}$. For $\gamma_k \leq \frac{1}{n \exp(r/\lambda)}$, the above inequality ensures that $\Delta_k$ is monotonically decreasing:

$$\Delta_{k+1} \leq \Delta_k - \left( \frac{2}{n \exp(r/\lambda)} - \gamma_k \right) \| \nabla L(\theta_k) \|^2 \leq \Delta_k.$$

To maintain $\theta_k \in B(r)$ for all $k$, choose $r$ such that

$$\| \theta_k \| \leq \Delta_k + \| \theta^* \|_{\Lambda_k^{-1}} \leq \| \Delta_0 \|_{\Lambda_k^{-1}} + \| \theta^* \|_{\Lambda_k^{-1}} \leq 2 \left( \| \theta_0 - \theta^* \| + \| \theta^* \| \right) = r.$$

This ensures that $\theta_k$ remains within the ball $B(r)$ for all iterations.

**Averaging.** Since $\theta_k \in B(r)$, we can take the average of (17) over $k = 1, \ldots, \ell$:

$$\sum_{k=1}^{\ell} \| \nabla L(\theta_k) \|^2 \leq n \exp(r/\lambda) \left( \sum_{k=1}^{\ell} \Delta_k - \Delta_{\ell+1} \right) \leq (n \exp(r/\lambda) + 1) \Delta_1 \leq (n \exp(r/\lambda) + 1) r.$$

**Gradient Convergence.** This leads to the following bound on the minimum gradient norm:

$$\min_{k \leq \ell} \| \nabla L(\theta_k) \|^2 \leq \tfrac{1}{\ell} \sum_{k=1}^{\ell} \| \nabla L(\theta_k) \|^2 \leq \tfrac{1}{\ell} (n \exp(r/\lambda) + 1) r. \tag{18}$$

**Closeness to Doubly Stochastic Matrices.** By definition,

$$\nabla L(\theta_k) = \begin{bmatrix} M^{(k)} \mathbf{1} - \frac{1}{n} \mathbf{1} \\ (M^{(k)})^\top \mathbf{1} - \frac{1}{n} \mathbf{1} \end{bmatrix}, \tag{19}$$

where $\mathbf{1}$ denotes the vector of all ones. Substituting the expression for $\nabla L(\theta_k)$ into (18) gives

$$\min_{k \leq \ell} \left( \| M^{(k)} \mathbf{1} - \frac{1}{n} \mathbf{1} \|^2 + \| (M^{(k)})^\top \mathbf{1} - \frac{1}{n} \mathbf{1} \|^2 \right) \leq \frac{(n \exp(r/\lambda) + 1) r}{\ell}. \tag{20}$$

## B  PROOF OF THEOREM 2

According to Thm. 1, a transformer can implement gradient descent. Therefore, the proof casts to analyzing gradient descent (with specific coordinate-wise stepsizes) on the convex $L$. However, we cannot directly apply existing convergence results from convex optimization. The existing convergence results for smooth convex optimization are in terms of function value $L$ when $L$ is not strongly convex[1]. But, the theorem statement aims at the convergence to the optimal transport map.

We take inspiration from convex optimization proof. The proof consists of two key steps: (i) the convergence of attention patterns to a matrix that is approximately doubly stochastic, and (ii) a hypothetical simulation of Sinkhorn's recurrence.

(i) As shown in Thm. 1, the transformer can perform gradient descent with an adaptive step size on the convex function $L$. Since $L$ is convex, gradient descent is guaranteed to converge to a stationary point where the gradient norm becomes zero. Specifically, it is straightforward to verify that $\nabla_u L = M \mathbf{1} - \frac{1}{n} \mathbf{1}$ and $\nabla_v L = M^\top \mathbf{1} - \frac{1}{n} \mathbf{1}$, where $M_{ij} = \exp \left( \frac{-C_{ij} + u_i + v_j}{\lambda} - 1 \right)$. Therefore, small gradients for $u$ and $v$ imply that $M$ is close to being doubly stochastic.

---

[1]It is easy to check that $L$ is not strongly convex since it Hessian has a zero eigenvalue.

(ii) We demonstrate that when the matrix $M$ is approximately doubly stochastic, it is near the desired solution $P_\lambda^*$. To establish this, we (hypothetically) run Sinkhorn's recurrence starting from $M$ and use its contraction property proven by Franklin and Lorenz (1989).

Before elaborating on the details of (i) and (ii), we present two propositions.

**Preliminaries.** Define the functions $row : \mathbb{R}_+^{n \times n} \to \mathbb{R}_+^n$ and $col : \mathbb{R}_+^{n \times n} \to \mathbb{R}_+^n$ as

$$\text{row}(A)_i = \frac{1}{n \sum_j A_{ij}}, \quad \text{col}(A)_j = \frac{1}{n \sum_i A_{ij}}.$$

We also introduce functions $f, g : \mathbb{R}^{n \times n} \to \mathbb{R}^{n \times n}$ defined as

$$f(A) = A\text{diag}(\text{col}(A)), \quad g(A) = \text{diag}(\text{row}(A))A.$$

Indeed, $f(A)$ (resp. $g(A)$) normalizes the columns (resp. rows) of $A$ by a scaling factor of their average. We will later use $f$ and $g$ to formulate Sinkhorn's recurrence, which iteratively normalizes the rows and columns of a matrix with positive elements. The next proposition proves that an almost doubly stochastic matrix remains almost doubly stochastic under $f$ and $g$. To formulate the statement, we introduce a set containing matrices that almost doubly stochastic matrices:

$$\mathcal{S}_\epsilon := \left\{ A \in \mathbb{R}_+^{n \times n} \mid \|A\mathbf{1}_n - \tfrac{1}{n}\mathbf{1}_n\|_\infty \leq \epsilon \text{ and } \|A^\top \mathbf{1}_n - \tfrac{1}{n}\mathbf{1}_n\|_\infty \leq \epsilon \right\}.$$

**Proposition 1.** *Suppose that $A \in \mathcal{S}_\epsilon$; then $f(A) \in \mathcal{S}_{3\epsilon}$ and $g(A) \in \mathcal{S}_{3\epsilon}$, as long as $\epsilon < 1/(3n)$.*

Recall the metric $d$ defined in (11). The next proposition establishes a particular property of $f$ and $g$ with respect to $d$.

**Proposition 2.** *Let $A \in \mathcal{S}_\epsilon$ be decomposed as $A = diag(w)Qdiag(q)$, where $w, q \in \mathbb{R}_+^n$.*

*(i) For $f(A) = diag(w)Qdiag(q')$, $d(q, q') \leq 4n\epsilon$ holds for $\epsilon < \frac{1}{4n}$.*

*(ii) For $g(A) = diag(w')Qdiag(q)$, $d(w, w') \leq 4n\epsilon$ holds for $\epsilon < \frac{1}{4n}$.*

**(i) Convergence Analysis.** According to Theorem 1, there is a choice of parameters such that

$$A_{ij}^{(\ell)} = e^{\frac{-C_{ij}+u_i^{(\ell)}+v_j^{(\ell)}}{\lambda}-1},$$

where $u^{(\ell)}$ and $v^{(\ell)}$ are the iterates defined in (4). The following lemma establishes that, as the number of iterations $\ell$ increases, $A^{(\ell)}$ meets a neighborhood of doubly stochastic matrices.

**Lemma 1.** *For $\gamma_k^{-1} = (n+2)e^{2r/\lambda}$, there exists a $k \leq \ell$ such that*

$$A^{(k)} \in \mathcal{S}_\epsilon, \quad \text{where } \epsilon^2 := \left(\frac{1}{\ell}\right) 3n e^{3r/\lambda} r.$$

Notably, the matrix $A^{(k)}$ has a specific structure that ensures $A^{(k)} \in \mathcal{S}_\epsilon$ is sufficient to approximate $P_\lambda^*$. To prove this statement, we leverage the contraction property of Sinkhorn's recurrence.

**Contractive Sinkhorn's Process.** According to the last lemma, there exists an iteration $k \leq \ell$ such that $A^{(k)} \in \mathcal{S}_\epsilon$. We then apply Sinkhorn's recurrence starting from $A_1 = g(A^{(k)})$ as:

$$A_{m+1/2} = f(A_m), \quad A_{m+1} = g(A_{m+1/2}).$$

Notably, we utilize the above recurrence solely for the proof; hence, there is no need for a transformer to implement this recurrence. According to the definition, $A_m$ can be decomposed as $\text{diag}(w_m)Q\text{diag}(q_m)$, where $Q_{ij} = e^{-C_{ij}/\lambda-1}$ and $q_m, w_m \in \mathbb{R}_+^n$. Sinkhorn (1967) proves that there exist unique vectors $w^*, q^* \in \mathbb{R}_+^n$ such that $P_\lambda^* = \text{diag}(w^*)Q\text{diag}(q^*)$, where $w_i^* = e^{u_i^*/\lambda}$ and $q_j^* = e^{v_j^*/\lambda}$. (Franklin and Lorenz, 1989) establish the linear convergence of $(w_m, q_m)$ to $(w^*, q^*)$:

$$\begin{cases} \mu(w_{m+1}, w^*) \leq \eta\mu(w_m, w^*) \\ \mu(q_{m+1}, q^*) \leq \eta\mu(q_m, q^*) \end{cases}, \quad \eta = \frac{\phi(A_1)^{1/2}-1}{\phi(A_1)^{1/2}+1} < 1, \tag{21}$$

where $\phi(A) = \max_{ijkl} \frac{A_{ik}A_{jl}}{A_{jk}A_{il}}$. Since $A_1 = \text{diag}(w_1)Q\text{diag}(q_1)$, we have $\phi(A_1) = \phi(Q)$.

**(ii) Approximating the Optimal Solution.** Propositions 1 and 2 enable us to demonstrate that there exists a constant $c$ such that $cA_1$ lies within a neighborhood of $P_\lambda^*$. Proposition 1 combined with Lemma 1 ensure $A_1 \in \mathcal{S}_{3\epsilon}$. Thus, we can apply Proposition 2 to obtain: $\mu(q_2, q_1) \le 12n\epsilon$. Using Proposition 1 again, we find that $A_{1+1/2} \in \mathcal{S}_{9\epsilon}$. Consequently, we can invoke Proposition 2 once more to yield: $\mu(w_2, w_1) \le 36n\epsilon$. Applying the triangle inequality together with 21 completes the proof:

$$36n\epsilon \ge \mu(w_2, w_1) \ge \mu(w_1, w^*) - \mu(w_2, w^*) \ge (1-\eta)d(w_1, w^*)$$
$$12n\epsilon \ge \mu(q_2, q_1) \ge \mu(q_1, q^*) - \mu(q_2, q^*) \ge (1-\eta)\mu(q_1, q^*).$$

## C    PROOF OF PROPOSITION 1

We prove $f(A) \in \mathcal{S}_{3\epsilon}$ and the proof for $g(A) \in \mathcal{S}_{3\epsilon}$ follows exactly the same. Since $A \in \mathcal{S}_\epsilon$, the following two inequalities hold

$$\left| \sum_i A_{ij} - \tfrac{1}{n} \right| \le \epsilon \implies \sum_i A_{ij} \ge \tfrac{1}{n} - \left| \tfrac{1}{n} - \sum_i A_{ij} \right| \ge \tfrac{1}{n} - \epsilon \tag{22}$$

Using the above two inequalities, we proceed as

$$\left| \tfrac{A_{ij}}{n\sum_i A_{ij}} - A_{ij} \right| = A_{ij} \left| 1 - \tfrac{1}{n\sum_i A_{ij}} \right| \tag{23}$$
$$\le \tfrac{A_{ij}\epsilon}{\sum_i A_{ij}} \tag{24}$$
$$\le \tfrac{A_{ij}\epsilon}{\frac{1}{n}-\epsilon}. \tag{25}$$

We use the above inequality to complete the proof:

$$\left| \sum_j \tfrac{A_{ij}}{n\sum_i A_{ij}} - \tfrac{1}{n} \right| \le \left| \sum_j \tfrac{A_{ij}}{n\sum_i A_{ij}} - \sum_j A_{ij} \right| + \left| \sum_j A_{ij} - \tfrac{1}{n} \right| \tag{26}$$
$$\le \sum_j \left| \tfrac{A_{ij}}{n\sum_i A_{ij}} - A_{ij} \right| + \epsilon \tag{27}$$
$$\le \tfrac{\epsilon}{1/n-\epsilon} \sum_j A_{ij} + \epsilon \tag{28}$$
$$\le \epsilon \left( 1 + \tfrac{1/n+\epsilon}{1/n-\epsilon} \right) \tag{29}$$

## D    PROOF OF PROPOSITION 2

We prove part (i), and the proof for part (ii) follows exactly the same. The following inequality holds for $A \in \mathcal{S}_\epsilon$:

$$\forall j : \left| \sum_i A_{ij} - \tfrac{1}{n} \right| \le \epsilon. \tag{30}$$

Using the above inequality, we get:

$$|q_j' - q_j| = \left| \tfrac{q_j}{n\sum_i A_{ij}} - q_j \right| \tag{31}$$
$$= q_j \left| \tfrac{1}{n\sum_i A_{ij}} - 1 \right| \tag{32}$$
$$\le q_j \left( \tfrac{\epsilon}{\frac{1}{n}-\epsilon} \right) \tag{33}$$

Plugging the above inequality into $\mu$ concludes the proof:

$$\tfrac{q_i q_j'}{q_j q_i'} \le \tfrac{1}{1-2n\epsilon} \implies \mu(q, q_i') \le \log(\tfrac{1}{1-2n\epsilon}) \le \tfrac{2n\epsilon}{1-2n\epsilon}. \tag{34}$$