# OpenReview forum: "Provable optimal transport with transformers: The essence of depth and prompt engineering"
_ICLR.cc/2025/Conference — Submitted to ICLR 2025_

### Official Review · Reviewer_YKZP · 2024-10-26

**Soundness:** 3
**Presentation:** 2
**Contribution:** 2
**Rating:** 5
**Confidence:** 3

**Summary:**

The paper proves the existence of a transformer which takes as an input a collection of points $(x_1,y_1, \dots, x_n,y_n),$ in $\mathbb{R}^d$ and outputs an approximate solution to the optimal transport matching problem between the sets of points. Their proof relies uses an explicit construction of a transformer which unrolls the gradient descent algorithm on the entropic-regularized OT problem for the dual potentials, and which takes a specifically-engineered prompt as an input. The convexity of the dual OT problem then implies a bound between the transformer output and the optimal coupling, which decreases as the depth of the transformer increases. Their result adds further evidence to the burgeoning interpretation of transformers as algorithm-learners.

**Strengths:**

1. The paper provides guarantees for a broad class of discrete OT problems, which arise in a variety of applied fields.
2. The main results are supported by rigorous proofs, which are correct to the best of my knowledge.

**Weaknesses:**

1. The idea of using transformers to unroll gradient descent algorithms is not new in the theory literature, it has been studied in [1], [2], and [3] extensively. In light of those results, it doesn't seem very surprising that you can also construct a transformer to unroll GD for the optimal transport loss.
2. The paper only bounds the approximation error of using transformers to solve OT problems and ignores other sources of error such as the pre-training generalization error and the training error of the transformer parameters.

Overall, I think the authors begin to tackle an interesting and important question in learning theory, but their analysis is not taken far enough.

[1] Bai, Yu, et al. "Transformers as statisticians: Provable in-context learning with in-context algorithm selection." Advances in neural information processing systems 36 (2024).
[2] Guo, Tianyu, et al. "How do transformers learn in-context beyond simple functions? a case study on learning with representations." arXiv preprint arXiv:2310.10616 (202
[3] Von Oswald, Johannes, et al. "Transformers learn in-context by gradient descent." International Conference on Machine Learning. PMLR, 2023.

**Questions:**

1. I think it would be helpful to comment on the choice of the adaptive step size in Theorem 1. My guess is that this is an artifact of using softmax attention, as it is difficult to express the gradients of the loss in terms of the softmax normalization. But it still seems strange that the step sizes depend on the parameters $u$ and $v$ which are being updated, as then it does not seem as though the transformer layers are really performing gradient descent. Does this affect the convergence of the transformer output to the true OT matrix?
2. In your experiments, how does the prediction of the trained transformer compare to the solution obtained by running gradient descent on the dual OT loss?
3. Is your specific prompt engineering important for the numerics? Clearly it plays a central role in the proof. However, I would expect that you could first apply a learnable feature map to the original prompt and then apply attention to the featurized prompt and obtain similar results (i.e, make the prompt engineering learnable).

---

> ### Author Response · Authors · 2024-11-22
> **Prompt engineering is effective in practice**
>
> **Questions:**
> > Is your specific prompt engineering important for the numerics? Clearly it plays a central role in the proof.
>
> Yes, please check added figure 4 in the revised document and **[G.2]** in the general response. The prompt engineering is indeed very important for solving optimal transport even for pretrained transformers. Thanks helping us to eleborate on this important aspect.
>
> > However, I would expect that you could first apply a learnable feature map to the original prompt and then apply attention to the featurized prompt and obtain similar results (i.e, make the prompt engineering learnable).
>
> Autmated engineering of prompt during training is the subject our follow-up in-progress research.
>
> > I think it would be helpful to comment on the choice of the adaptive step size in Theorem 1.
>
> You are right: softmax limits the choice of stepsize. However, we prove there is a choice of stepsize that allows provable convergence in Theorem 3.
>
> >  it does not seem as though the transformer layers are really performing gradient descent.
>
> This is a variant of gradient descent with adapative coordinate-wise stepsizes. Adam and Adagrad are examples of gradient descent with coordinate-wise apative stepsizes.
>
> >  Does this affect the convergence of the transformer output to the true OT matrix?
>
> Yes. Theorem 3 proves this particular adaptive stepsize can still acheive a convergence rate. Notably, some part of engineered prompt are carefully chosen to obtain adaptive stepsizes that ensure the convergence.
>
> > In your experiments, how does the prediction of the trained transformer compare to the solution obtained by running gradient descent on the dual OT loss?
>
> We do not claim that this specific stepsize choice comes with empirical benefits. However, the implementation of gradient descent was more challenging for softmax attentions due to the normalization in softmax. We added a paragraph in discussion to motivate future studies on this line of research (see Depth Efficient Guarantees in Discussions).
>
>
>
>
> **Weaknesses:**
> > The idea of using transformers to unroll gradient descent algorithms is not new in the theory literature, it has been studied in [1], [2], and [3] extensively.
>
> Please check general response **[G.1]**
>
> > The paper only bounds the approximation error of using transformers to solve OT problems and ignores other sources of error such as the pre-training generalization error and the training error of the transformer parameters.
>
> We added figure 3 to eleborate on the role of training. Please check the general respone **[G.3]**.

---

> > ### Comment · Reviewer_YKZP · 2024-11-26
> > **Reply to author**
> >
> > I thank the authors for their thorough response to my questions. My question about the role of the softmax has been made clear and think the construction of a transformer to unroll gradient descent is more novel than I originally thought, since previous works use either linear attention or a pointwise nonlinearity in place of the softmax. I will maintain my score, as I still feel the paper is limited since its main contribution is an existence result.

---

> > > ### Author Response · Authors · 2024-11-26
> > > **Thanks for confirming the novelty**
> > >
> > > Thank you so much for confirming that "... the construction of a transformer to unroll gradient descent is more novel than I originally thought, since previous works use either linear attention or a pointwise nonlinearity in place of the softmax ...".

---

### Official Review · Reviewer_mAff · 2024-11-01

**Soundness:** 4
**Presentation:** 1
**Contribution:** 2
**Rating:** 6
**Confidence:** 4

**Summary:**

The paper addresses the problem of theoretical guarantees of solving the entropic optimal transport (EOT) in the context of Wasserstein-2 distance using transformers. The authors provide theoretical assurance by demonstrating that transformers when equipped with specific prompts perform gradient descent on the dual formulation of the EOT problem.  The key contributions include proving that transformers with fixed parameters can implement gradient descent on the Wasserstein-2 EOT problem and establishing the error bound for this approach. The experimental results further demonstrate the ability of transformers with specific prompts to sort lists of numbers supporting the theoretical findings.

**Post rebuttal**: Score increase 3->6, see the discussion.

**Strengths:**

The paper is rigorously constructed, demonstrating strong theoretical foundations, and supporting their claims of transformers being able to conduct gradient descent on optimal transport problems with explicit error bounds and convergence metrics. While there is a lack of available experiments specifically related to optimal transport, the authors find a way around this limitation by conducting experiments on sorting and optimal transport for different input sizes, which validate the theoretical assertions and strengthen the quality and reliability of the paper's contributions.

**Weaknesses:**

**Missing Generalization**. The authors establish a link between transformers and optimal transport, providing a specific and *very artificial* construction. While the choice of parameters, architecture, and prompts is justifiable within this framework, the construction remains limited, as there is no exploration beyond these specific Optimal Transport (OT) costs or an attempt to generalize the findings. Different applications and data types may require distinct cost functions to accurately reflect the underlying transport dynamics; for instance, a Manhattan cost better represents routing scenarios in transportation logistics, while a Gaussian cost can effectively model probabilistic asset movements in finance. By restricting their focus to specific OT costs, the authors miss potentially new application insights in the research field.

**Limited Scope**. Numerous studies [1], [2], [3] are bridging the gap between the capabilities of transformer models in artificial scenarios and their performance in real-world learning tasks, which is crucial for a comprehensive understanding of in-context learning. While the authors successfully establish a connection between transformers and optimal transport, they neglect to address whether these models are genuinely capable of solving optimal transport problems when trained in real-world scenarios. This omission limits the depth of their analysis and may hinder the reader's ability to fully grasp the implications of their findings within practical contexts. A discussion on this point would significantly enhance the paper’s contribution to the field.

[1] Von Oswald, Johannes, et al. "Transformers learn in-context by gradient descent." International Conference on Machine Learning. PMLR, 2023.

[2] Akyürek, Ekin, et al. "What learning algorithm is in-context learning? investigations with linear models." arXiv preprint arXiv:2211.15661 (2022).

[3] Wang, Jiuqi, et al. "Transformers Learn Temporal Difference Methods for In-Context Reinforcement Learning." arXiv preprint arXiv:2405.13861 (2024).

**Mistakes in Equations**. The equations contain several errors that make it difficult to follow the logical flow of the proofs. Some examples include:
- at line 241 the definition of $\lambda Q^{(l, 1)}$ contains excessive $\mathbf{0}{d'}$ vector at the end;
- at line 241 and 242 vectors $e{2d+5}$ in both definitions of $\lambda Q^{(l, 1)}$ and $\lambda Q^{(l, 2)}$  must be moved more left to get additional term $v^{(l)}$ or $u^{(l)}$ in the equation at line 291;
- at line 275 the iteration index of $Z$ has to be $(l+1)$;
- at line 286 some of the zeros must be zero vectors or matrix;
- at 291 the $\lambda$ is missing and, additionally, the expression for the upper left block is wrong;
- at line 304 there must not bee coefficient $\gamma$, instead it should be moved to line 309.

These errors disrupt the coherence of the mathematical arguments and create unnecessary confusion. Carefully revising each equation to ensure consistency and accuracy would significantly enhance readability and strengthen the rigor of the presentation.

**Late or Missing Introduction of Notation or Definitions**. Notation is often introduced at various points of the paper, making it challenging for readers to locate definitions or fully understand terms when they first appear. In some cases, notation seems to be used without any prior introduction, leaving readers to infer meanings from context, which reduces the accessibility of the theoretical sections. It was mentioned:

- at line 202 the meaning of the subscript of $[Z^{(l)}_n]$ is not defined clearly;
- at line 299 it is not seen that this matrix is attention matrix $A$;
- at line 286 and 291 is used different notation of $[||x_1||^2, \dots, ||x_n||^2]$, $||x||^2$ and $x^2$;
there is no clear evidence of whether column or row vector notation is used;
- at line 237 authors introduce $d' = 2d + 9$ but they still use $2d + 9$ in lines 267, 275.

*Suggestion:* A more effective approach would be to introduce all key notations systematically in a dedicated section at the beginning of the paper. Alternatively, providing a table summarizing all notations and definitions, with clear explanations and references to where they are first used, would help readers navigate the mathematical content more easily.

**Difficulty in Understanding Matrix Derivations.** In several parts of the paper, it is challenging to follow the derivation, for example, in lines 291 or 308. The transitions from definitions to their matrix forms are often abrupt, leaving readers uncertain about how certain matrices are constructed or why they take particular forms. In several instances like in lines 291, 304, 309, and 314, the dimensions of matrices are not specified, which is crucial for notation with blocks.

*Suggestion:* To improve clarity, consider adding more intermediate steps in the Appendix. Providing explicit dimensions for each matrix upon introduction would make it easier for readers to follow matrix operations. Including dimensions directly in matrix definitions or accompanying explanations would also help readers track the flow of derivations and verify the correctness of matrix manipulations.

**Questions:**

To strengthen the paper, please consider:

- consider bridging the gap between whether transformers can perform gradient descent on optimal transport tasks and whether they
- conduct it in real-world scenarios;
- consider different OT  costs;
- adding more intermediate steps of matrix derivations in the Appendix;
- including dimensions directly in matrix definitions and accompanying explanations;
- extending notation section with more notations that are used in main sections;
- fixing mentioned mistakes;
- including a section that explicitly discusses the relevance of optimal transport theory to transformer models, explaining how this perspective can lead to new insights into the mechanics.

---

> ### Author Response · Authors · 2024-11-22
> **"Missing generalization" may be the limitation of transformers**
>
> > "Missing Generalization": using a specific cost function for optimal transport
>
> This is not a limitation of our result, but it may be the limitation of transformers used in practice. Transformers have their own limitations and can not solve all problems. Focusing on $\ell_2$ cost for optimal transport is due the fact that we were only able to prove that transformers can solve optimal transprot for this specific cost function. Given $x_1$ and $y_1$, how can we consturct an attention layer that simply computes $\|x_1 - y_1\|_1$?  It is not clear whether transformers can provably solve optimal transport with a cost function different from from norm 2.  We clearly specify the cost in the abstract to avoid confusions about the choice of cost function for optimal transport.
>
> To solve optimal transport with a differnt cost, we beleive the prompt must contain $C$ where $C_{ij} = cost(x_i,y_j)$ in instead of points $x_i$ and $y_i$. The main issue with this formulation is this formulation is beyond standard prompting. Consider example of sorting. We can give an unsorted list containing $x_1, \dots, x_n$ to transformers to sort. But, consturcting matrix $C_{ij} = cost(x_i,y_j)$ from $x_i$ and $y_i$ is not a standard prompting.
>
>
>
> > Limited Scope. Numerous studies [1], [2], [3] are bridging the gap between the capabilities of transformer models in ...
>
>
> Please check the general response **[G.1]**.
>
>
>
> > Mistakes in Equations
>
>
> We appreciate your careful check of the proof. There is no  mistake in our proof and there are only two typos.
>
> > at line 241 the definition of contains excessive vector $0$
>
> This extra vector is needed for experiments in Figures 3 and 4 in the revised pdf. This is not a mistake but was a leveraged degree of freedom.
>
> > at line 241 and 242 vectors in both definitions of  and
>  must be moved more left to get additional term  or in the equation at line 291;
>
> There is no need for shift to left.
>
> > at line 275 the iteration index of  has to be;
>
> This typo is fixed.
>
> > at line 286 some of the zeros must be zero vectors or matrix;
>
> We omit matrix/vector notions when it can be implied from the context.
> > at 291 the is missing and, additionally, the expression for the upper left block is wrong;
>
> This is only a typo fixed in blue in the revised version. Notably, this typo does not change any following derviations.
> > at line 304 there must not bee coefficient, instead it should be moved to line 309.
>
> Notably, matrix $D_\ell$ contains $\gamma_\ell$.
>
> > Difficulty in Understanding Matrix Derivations.
>
> We will improve the presentation and apply your comments in the camera-ready version. Remarkably, matrix derivations are not the main components of our proof. Details on matrices dimensions can be concluded from the context. We beleive that all the reviewers were able to follow the proofs. For example, reivewers LUYE assess "The paper generally communicates complex ideas effectively, with clear explanations of both the background and the transformer mechanics used in OT tasks.". Furthermore, reviewer and YKZP confirm "The main results are supported by rigorous proofs...".
>
>
> **Questions**
>
> > including a section that explicitly discusses the relevance of optimal transport theory to transformer models, explaining how this perspective can lead to new insights into the mechanics.
>
> One key insight that we bring is the possibility of **out-of-distribution generalization** with transformers. Suppose that we train a transformer to sort lists of size $7$. Is it possible to use the same transformer to sort lists of size $9$.  Please check added *figure 3* in the revised pdf to see this striking capability. This is an example of out-of-distribution generalization. In this paper, we take a step towards explaining this generalization, showing a transformer can sort lists of different sizes.
>
> Interestingly, we use a specific **prompt engineering** for the proof. While prompt engineering is widely used for langauge models, it is not well understood how it can improve the performance of language models. Here, we prove that prompt engineering can enrich the computation power of transformers allowing them to solve fundamental optimal transport problem. Please check revised *figure 4* showing the practical benefits of the proposed engineered prompt.  This insight can be greatly leverage in future theoretical and practical studies.
>
> > consider bridging the gap between whether transformers can perform gradient descent on optimal transport tasks and whether they
> conduct it in real-world scenarios
>
> Please check the general response **[G.3]**.
>
> > consider different OT costs;
>
> We question the possibility of solving OT with a general cost function with standard transformers. As explained above, transformers have their own limitations.

---

> > ### Comment · Reviewer_mAff · 2024-11-29
> > **Response**
> >
> > Thank you for addressing my concerns and explaining the focus on the $\ell_2$ cost and its role in the framework. Your detailed responses and new experiments, which clarify the choice of parameters and the role of prompt engineering, significantly strengthen the theoretical foundation. I will raise my score to acknowledge the effort in bridging the gap between what transformers can theoretically do and what they achieve in practice. However, the main novelty is in extending the theory to include non-linear softmax attention and demonstrating that transformers can handle varying sample sizes while solving the target task. Based on the answers and the revision, I decided to increase my score to 6. I still think this is a borderline paper and will wait to hear the opinion of the other reviewers.

---

> > > ### Author Response · Authors · 2024-11-29
> > > **Thank you**
> > >
> > > We greatly appreciate your detailed, constructive feedback, which helped us improve the presentation. Thank you for taking the time to review the revision during the thanksgiving holidays.

---

### Official Review · Reviewer_LUYE · 2024-11-02

**Soundness:** 3
**Presentation:** 2
**Contribution:** 3
**Rating:** 5
**Confidence:** 2

**Summary:**

This investigates whether transformers can provide theoretical guarantees in solving the Sinkhorn optimal transport problem. The authors demonstrate that transformers, equipped with appropriately engineered prompts, can approximate optimal transport solutions by leveraging attention layers and simulating gradient descent with adaptive step sizes.

The authors designed prompts to aid transformers in solving optimal transport by providing the required memory and statistics within the model's layers. This setup helps in implementing gradient descent and enables transformers to handle various problem instances efficiently.

The study finds that the performance of transformers in solving optimal transport improves with increased depth, establishing an explicit approximation bound that converges as the model’s depth increases.

Sorting is shown as a special one-dimensional case of optimal transport, where transformers can approximate the sorted output by solving the transport problem. The experiments validate the theoretical predictions, showing that transformers can effectively approximate sorted sequences.

The paper rigorously proves that attention matrices in transformers can converge to a doubly stochastic matrix that approximates the optimal transport solution. This convergence avoids rank collapse, a common issue in attention layers, thus preserving the expressiveness of transformers.

**Strengths:**

Originality

The paper is original in its approach to connecting transformers with provable guarantees in solving the Sinkhorn OT problem. It leverages the attention mechanism of transformers in an innovative way, showing that transformers can approximate optimal transport solutions under entropic regularization.

Quality

The paper is rigorous in both its mathematical formalism and experimental validation. The theoretical development is well-supported by proofs, such as the gradient descent simulation using attention layers and the convergence analysis of attention matrices.

Clarity

The paper generally communicates complex ideas effectively, with clear explanations of both the background and the transformer mechanics used in OT tasks. All proofs are included in the main paper.

Significance

Theoretically, it provides new insights into how transformers can function beyond sequence modeling, linking them to iterative optimization methods such as gradient descent. This connection strengthens the iterative inference hypothesis, suggesting transformers can perform complex tasks in optimization by layer-wise refinement.

**Weaknesses:**

1. The paper's theoretical proofs are presented with a high level of mathematical density that may hinder accessibility for a broader audience, particularly readers less familiar with optimization theory or transformers' inner workings. To improve accessibility, the authors could add intermediate explanations or visual aids (such as flowcharts or diagrams) to clarify the key steps in the proofs. An appendix with expanded explanations of some technical choices and motivations, particularly around the parameter settings in Section 4, could further aid comprehension.

2. Prompt engineering is a significant aspect of the paper’s contribution, especially given its role in facilitating the transformer’s ability to approximate the optimal transport problem. However, the authors do not fully explore the sensitivity or robustness of their prompt design. For example, it is unclear if minor variations in the prompt structure or parameters would significantly impact the model's performance in solving OT, which is crucial for understanding the generalizability of this approach. Adding a sensitivity analysis that examines the impact of slight prompt modifications on performance would strengthen the prompt engineering component.

3. The experiments are relatively limited, focusing primarily on simple sorting tasks. For example, demonstrating performance on high-dimensional OT tasks, even with a modest dimensionality (e.g.,  𝑑=10), would provide valuable insights into the approach's scalability and robustness.  A head-to-head comparison with other OT algorithms (e.g., Sinkhorn or linear programming methods) on shared metrics (e.g., accuracy, runtime, or scalability) would provide a clearer picture of where the transformer-based approach stands.

4. This paper focuses on using transformers for OT.  How this framework could extend to broader optimization or machine learning problems?

**Questions:**

see weaknesses.

---

> ### Author Response · Authors · 2024-11-22
> **Prompt engineering is essential in theory and practice**
>
> > Prompt engineering is a significant aspect of the paper’s contribution ...
>
> As mentioned in the general response **[G.2]**, we have added the experimental results in Figure 4, which substantiate the important role of prompt engineering. In this experiment, we compare the performance of trained transformers with and without prompt engineering for solving optimal transport. We observed that the engineered part of the prompt (highlighted in red in Equation 6) significantly enhances the performance of transformers.
>
>
>
> > The paper's theoretical proofs are presented with a high level of mathematical density that may hinder accessibility for a broader audience
>
> The choice of parameters/prompt is based on the subtle observation that gradient descent with adaptive coordinate-wise step sizes on $L(u,v)$ relies on the computation of the matrix $M_{ij} = e^{\left( \frac{-C_{ij} + u_i + v_j }{\lambda} - 1 \right)}$ as $\nabla_u L = 1_n M - \frac{1}{n} 1_n$. This computation can be performed by a single attention layer.
>
> Recall that softmax attention consists of three main computations:
>
> - **(i)** Computing inner products of tokens.
> - **(ii)** Computing element-wise exponentials of inner products.
> - **(iii)** Normalizing each row of the matrix.
>
> Using **(i)** and **(ii)**, along with including $u$ and $v$ in the engineered prompt, we construct $M$ in Equation 8.
>
> In Equation 8, you will notice additional $1$ and $0$ elements after steps **(i)** and **(ii)**. These elements allow us to derive the coordinate-wise step sizes after applying the normalization in softmax (**iii**). Notably, the step sizes are tuned so that we can achieve a convergence rate as shown in Theorem 2.
>
> We use colors in the proof to highlight different components of the engineered prompt, which enable us to compute the gradient descent on $L$. We will include the above summary in the appendix or main text.
>
> > This paper focuses on using transformers for OT. How this framework could extend to broader optimization or machine learning problems?
>
>
> We demonstrate that prompt engineering can significantly boost the computational performance of transformers. This aligns with experimental observations, which show that adding sentences to the prompt often leads to improved responses. We added experiments in Figure 4 to further eleborate on the important role of prompt engineering. Our study highlights the need to revisit the role of prompt engineering in enhancing the computational power of transformers.
>
> This study serves as a foundational step in a larger framework that establishes the computational power of language models. This framework connects transformers to optimization methods, thereby proving that transformers can solve a variety of problems, including regression and policy evaluation in Reinforcement Learning. While current literature predominantly relies on oversimplified models, such as linear attention, our results hold for softmax attention, which is widely used in practice. We show that linking transformers to optimization methods is effective, even for transformers deployed in real-world applications. We highlight prompt engineering can be leverage to boost the computational prower of transformers.
>
> > The experiments are relatively limited
>
> We add two experiments in the revision: (i) experiments on trained transformers in *figure 3* (ii) experiments showing the siginficant of the engineered prompt in *figure 4*. We will add experiments for $d>1$ in future revisions.

---

> > ### Author Response · Authors · 2024-12-01
> > **Follow-Up on Rebuttal Feedback**
> >
> > Dear reviewer,
> >
> > Thank you very much for your constructive feedback. One of the reviewers increased their score by three points after reading our rebuttal, suggesting that we have adequately addressed the primary concerns. If you have any additional questions or require further clarification, please let us know.
> >
> > We sincerely appreciate your time and consideration.

---

> > > ### Author Response · Authors · 2024-12-03
> > > **Request for considering our response**
> > >
> > > Dear reviewer,
> > >
> > > We appreciate it if you consider our response in your evaluation and final decision.
> > >
> > > Thank you very much

---

### Official Review · Reviewer_XEkW · 2024-11-04

**Soundness:** 3
**Presentation:** 3
**Contribution:** 2
**Rating:** 5
**Confidence:** 4

**Summary:**

This article constructs a prompt that enables transformers to perform gradient descent on dual optimal transport. Additionally, it provides an explicit approximation bound that improves with increasing depth. These results offer guarantees for the effectiveness of transformers in solving the optimal transport problem.

**Strengths:**

This paper introduces a novel inquiry into whether transformers can effectively implement optimal transport. Additionally, the construction is independent of the number points, and can be used to solve optimal transport problems of different scales.

**Weaknesses:**

- The experiments in Section 6 only evaluate constructed transformers. However, it remains unclear whether training standard transformers can achieve the solutions constructed in the theory.

- Lines 375 to 384 in Section 5 and lines 479 to 483 in Section 6 imply that attention patterns maintain full rank. This assertion contradicts to the commonly observed phenomenon in practice, where attention matrices are highly low-rank and sparse.

- The proof is based on the constructuion, so the required depth is only a sufficient condition. The required depth is presented as a sufficient theoretical condition but does not preclude the possibility that shallower transformers may also effectively perform optimal transport.

- Although this paper focuses on a novel task (optimal transport), the primary proof idea is similar to that of many previous works: using multi-layer transformers to simulate multi-step GD iterations.

**Questions:**

- Could the authors construct transformers with low-rank and sparse attention matrices, which can also effectively implement optimal transport?
- I am interested in the insights derived from the prompt construction, and the paper would benefit from a more detailed discussion on this topic.

---

> ### Author Response · Authors · 2024-11-22
> **Prompt engineering is befentifial in practice and can avoid the rank collapse of attentions**
>
> **Questions.**
>
> > I am interested in the insights derived from the prompt construction, and the paper would benefit from a more detailed discussion on this topic.
>
> We have added Figure 4 to elaborate on the role of prompt engineering. We observed that training without prompt engineering results in poor performance on optimal transport. Furthermore, attention layers remain low rank after training. Thank you for helping us develop this insight.
>
>
> > Could the authors construct transformers with low-rank and sparse attention matrices, which can also effectively implement optimal transport
>
> We beleive low-rank attention layer can not solve optimal transport since the solution of optimal transport is a full-rank permuation matrix.
>
> **Weaknesses.**
> > The experiments in Section 6 only evaluate constructed transformers.
>
> Please refer to the general response **[G.1]**. We have added experiments to further elaborate on (i) the mechanism of trained networks in Figure 3, and (ii) the significant role of prompt engineering in training:
>
> - **(i)** Trained transformers can sort larger lists after being trained on smaller ones, similar to the constructive transformer (see **[G.3]** in the general response).
> - **(ii)** Without the engineered prompt, transformers are unable to learn to sort in practice.
>
>
>
> > ... imply that attention patterns maintain full rank. This assertion contradicts to the commonly observed phenomenon in practice, where attention matrices are highly low-rank and sparse.
>
> The attention pattern remains **sparse** since the solution of optimal transport approximates a permutation matrix (see Figures 2 and 3).
>
> One of our main contributions is proving that attention layers can maintain a **high rank**. We have added a new experiment in Figure 3 to demonstrate that even trained transformers maintain a high rank. Figure 4 presents a notable observation: without prompt engineering, the attention pattern does not maintain a high rank. This experiment suggests that specific **prompt engineering**, combined with the choice of parameters, enables attention layers to maintain a **high rank** across layers.
>
>
> > The proof is based on the constructuion, so the required depth is only a sufficient condition ... does not preclude the possibility that shallower transformers may also effectively perform optimal transport.
>
> This limitation applies to shallow transformers, not to our theory. We observe that the performance of transformers improves with the addition of layers, as shown in Figure 3.
>
> Increasing depth is an effective strategy to enhance the performance of language models. For example, GPT-2 XL, with 48 layers, achieves better performance compared to GPT-2, which has 12 layers. Providing insights into the importance of depth is one of our key contributions.
>
> Related literature also confirms that deeper networks are computationally more powerful, including references [1-6] in the general response.
>
>
> > Although this paper focuses on a novel task (optimal transport), the primary proof idea is similar to that of many previous works: using multi-layer transformers to simulate multi-step GD iterations.
>
> We compared our results to the related literature in the general response **[G.1]**. What sets our results apart are:
>
> - **(i)** Using attention layers with softmax rather than linear attentions.
> - **(ii)** Demonstrating that prompt engineering can significantly boost the computational power of transformers for optimal transport (see Figure 4).
> - **(iii)** Showing that transformers can solve problems of varying sizes.

---

> > ### Author Response · Authors · 2024-12-01
> > **Follow-up**
> >
> > Dear reviewer,
> >
> > We believe that our response adequately addresses major concerns as we added experiments in the revision.
> > If you have any additional questions or concern, please let us know.
> >
> > Thank you very much for your time and consideration.

---

> > > ### Comment · Reviewer_XEkW · 2024-12-02
> > >
> > > Thanks for the clarifications and additional experiments. However, I find that the current explanations do not address my primary concerns, and I will maintain my initial score. Specifically:
> > >
> > > - It remains unclear whether the solution constructed by this work corresponds to the solution that standard transformers would converge to when trained on the optimal transport task.
> > >
> > > - While it is evident that increasing network depth can improve performance in practive, I am concerned about the necessity of the required depth in the work. The construction relies on a large depth, yet I question whether such depth is necessary for this specific task.
> > >
> > > - The experiments primarily emphasize the necessity of prompt word engineering but fail to provide high-level insights into the underlying mechanisms.
> > >
> > > - Although the authors clarified the technical distinctions from related works, I still believe that the core idea of the proof closely follows these works: using multi-layer transformers to simulate multi-step gradient descent iterations. Consequently, the presented results lack sufficient novelty (,even if task-specific constructions are required).

---

> ### Author Response · Authors · 2024-12-02
> **Many papers are based on unfolding gradient descent**
>
> Thank you so much for taking time and reviewing our response.
>
> >While it is evident that increasing network depth can improve performance in practive, I am concerned about the necessity of the required depth in the work. The construction relies on a large depth, yet I question whether such depth is necessary for this specific task.
>
> Please check paragraph *(i) Depth Efficient Guarantees* in Discussions. We clearly explained that our convergence bound may not be tight and transformers may need less depth in practice. To our knowledge, there is no baseline or existing results that proves transformers achieve an approximation error with a certain number of layers. Thus, we can not compare with other bounds and assess whether our bound is tight. Yet, we establish the **first bound** and future studies can improve it.
>
>
> > The experiments primarily emphasize the necessity of prompt word engineering but fail to provide high-level insights into the underlying mechanisms.
>
> The high level insight is that it may not be possible to unfold gradient descent iterations without prompt engineering. To implement gradient descent on $L$, we need to compute $C_{ij} = \|| x_i - y_j\||^2$ and store iterates $u_\ell$ and $v_\ell$. We engineered prompt to realize computing $C_{ij}$ using attention recurrence. Without prompt engineering, the transformer may not be expressive enough to implement gradient descent.  Indeed, using prompt engineering to implement gradient descent on particular objective sets our results apart from other works that rely on implementing gradient descent with transformers.
>
>
> > Although the authors clarified the technical distinctions from related works, I still believe that the core idea of the proof closely follows these works: using multi-layer transformers to simulate multi-step gradient descent iterations. Consequently, the presented results lack sufficient novelty (,even if task-specific constructions are required).
>
> There are many papers published and frequently cited that all rely on the idea of unfolding gradient descent with transformers layers, including references [1]-[8] in the general response. 4 references among these papers focus on the same task of linear regression. Still, these papers are published due to novelty and their distinguished settings and results. Using similar proof techniques is standard method in research and we believe it does not reduce the novelty. For example, statistical learning theory is based on proof techniques from empirical process theory, which itself heavy rely on concentration bounds in probability theory.
>
> Again, we stress that we used softmax attention for a task completely different from [1]-[8]. Non of these papers are actively leveraging from prompt engineering.
>
> > It remains unclear whether the solution constructed by this work corresponds to the solution that standard transformers would converge to when trained on the optimal transport task.
>
> We do not claim that we analyze trained transformers. We prove that transformers with prompt engineering can provably solve optimal transport on various input sizes.
>
> Our results can provide insights on the mechanism of trained transformers. Consider the result in Figure 3 showing a transformer trained to solve sorting on lists with length 7 can sort lists of length 9. The question that we address is: How such striking generalization is even possible given that the distribution of inputs are completely different due to their length difference? Here, we prove this generalization is possible due to the implementation of gradient descent.
>
> Notably, there are many different weight configurations that can implement the same algorithm (gradient descent with adaptive stepsizes). Thus, we can not compare the structure of weight matrices to conclude whether the algorithm implemented by trained transformers is gradient descent. However, we see that the process of solving optimal transport is also iteratively across the layers of transformers in Figure 3.
>
> Thanks again for considering our response. Please let us know if we can provide further clarification.

---

### Author Response · Authors · 2024-11-22
**General response**

We thank reviewers for their encouraging positive assessments, including
- "The paper is rigorous in both its mathematical formalism and experimental validation." by reviewer *LUYE*
- "The main results are supported by rigorous proofs" by reviewer *YKZP*
- "The paper is rigorously constructed, demonstrating strong theoretical foundations" by reviewer *mAff*.

Your constructive reviews greatly help us to improve the paper quality of experiments and elaborate on the important role of prompt engineering.

Bellow, we address general concerns about our results.

### **[G.1] Reviewers were concerned about the related literature on in-context learning [1-6]**

- Theoretical results for in-context learning of regression are often restricted to overly simplified cases using *linear attention* without softmax [1,3-6]. In contrast, our theory applies to **non-linear softmax attention**, which is commonly used in practice. In fact, softmax is crucial for our proof.
- In theory, **prompt engineering** plays a critical role in our established proof. To demonstrate the practical advantages of the engineered prompt, we have included additional experiments in Figure 4 of the revised submission. In this figure, we show that removing the engineered component of the prompt significantly reduces the performance of transformers in solving optimal transport. To the best of our knowledge, prompt engineering is not required for other in-context tasks like regression [1-5] or reinforcement learning [6].
- We have proven that transformers can handle optimal transport across **varying sample sizes**, whereas the literature [1,3-6] typically fixes the sample size. Our results advance the understanding of out-of-distribution (length) generalization, as illustrated in the revised Figure 3.
- Our primary focus is on solving optimal transport, whereas the studies [1-6] mainly concentrate on solving linear regression [1-5]. Importantly, implementing gradient descent for ridge regression does not imply that transformers can implement gradient descent for other functions required to solve optimal transport. While pre-trained transformers on natural language tasks often fail to solve linear regression [7], they frequently succeed in tasks like list sorting, which is a specific case of optimal transport.




### **[G.2] The Role of Prompt Engineering in Practice**

We have added experiments in the revised submission (Figure 4) demonstrating that the engineered prompt significantly enhances the performance of transformers in solving optimal transport tasks.



### **[G.3] Constructive Proof and the Role of Pretraining**

We have added experiments (Figures 3 and 4) to assess the role of pretraining in optimal transport. Figure 3 shows that pre-trained networks share similar properties with networks that use constructive weights and provably solve optimal transport:
- Both types of transformers iteratively solve optimal transport across layers.
- Both demonstrate the capability of **length generalization**, as they can sort lists of varying sizes.

Our results are a first step towards understanding the mechanism of optimal transport with transformers and explaining these properties. Notably, all the papers cited by reviewers [1-6] focus on constructive approaches. Constructive results have been fundamental to research on the mechanisms of pre-trained transformers for in-context learning. For example, [8] heavily relies on the constructive proof from [1] to demonstrate that pre-trained linear transformers can implement gradient descent on regression.



**References**

[1] Von Oswald, Johannes, et al. "Transformers learn in-context by gradient descent." International Conference on Machine Learning. PMLR, 2023.

[2] Akyürek, Ekin, et al. "What learning algorithm is in-context learning? investigations with linear models." arXiv (2022).

[3] Bai, Yu, et al. "Transformers as statisticians: Provable in-context learning with in-context algorithm selection." Advances in neural information processing systems 36 (2024).

[4] Guo, Tianyu, et al. "How do transformers learn in-context beyond simple functions? a case study on learning with representations." arXiv

[5] Von Oswald, Johannes, et al. "Transformers learn in-context by gradient descent." International Conference on Machine Learning. PMLR, 2023.

[6] Wang, Jiuqi, et al. "Transformers Learn Temporal Difference Methods for In-Context Reinforcement Learning." arXiv  (2024).

[7] Shen, Lingfeng, Aayush Mishra, and Daniel Khashabi. "Do pretrained Transformers Really Learn In-context by Gradient Descent?." arXiv  (2023).

[8] Ahn, Kwangjun, et al. "Transformers learn to implement preconditioned gradient descent for in-context learning." NeurIPS (2023).

---

### Meta-Review · Area_Chair_37MM · 2024-12-21

**Metareview:**

In this paper, the authors show that there exists a particular prompt with a specific configuration of parameters for transformers that implement Sinkhorn's algorithm and thus approximatively solve regularized optimal transport problems.

Having looked at the paper and the reviews, I see two main weaknesses in this work:

1. The experimental section could be improved: the authors do not address the question of whether the considered transformers are practically capable of solving optimal transport problems when trained in real-world data. At this point, it is not clear if the "out-of-distribution generalization" mentioned by the authors would occur in practice. Showing that transformers can practically learn the stinkhorn's algorithm would be a significant contribution (even on toyish data).

2. Lack of clarity and mistakes in equations: The paper still contains many typos and lacks some definitions after the revision. Actually, I am not sure the authors implemented what they claimed to in the rebuttal (e.g., the many comments of Reviewer mAff)

Although this paper's idea and result are interesting, I will recommend rejection for the above reasons. However, I believe the idea and the result have high potential, and I encourage the authors to address the abovementioned points in future submissions.

**Additional Comments On Reviewer Discussion:**

Reviewers were on the fence. Ultimately, I believe that the cons outweighed the pros for this paper.

---

> ### Public Comment · ~Hadi_Daneshmand1 · 2025-05-01
> **Misunderstandings in the Meta-Review**
>
> This meta-review contains major misunderstandings that reflect a fundamental misinterpretation of both the results and the reviews:
>
> - The paper does not claim that transformers implement Sinkhorn's algorithm, as incorrectly stated in the meta-review. Instead, it proves that transformers can implement gradient descent with adaptive step sizes on the dual Lagrangian function.
>
> - Figure 3 clearly demonstrates that out-of-distribution generalization occurs in practice after training which is stated as the main weakness in the meta review.
>
> - Although reviewer mAff raised their score from 3 to 6 after the rebuttal, the meta-review inaccurately claims that the response did not address the reviewer’s concerns.

---

> > ### Comment · Area_Chair_37MM · 2025-05-01
> > **Apologizes and Clarification**
> >
> > Dear Hadi,
> >
> > First, I apologise for my mistake. I agree that the claim of the paper is that "approximatively solve regularized optimal transport problems" (as mentioned in my Meta review) not via Sinkhorn's algorithm but via implementing gradient descent on the dual Lagrangian. However, this rectification does not affect the spirit of my meta review.
> >
> > Before giving a detailed answer, I would like to mention that I take my role as a meta-reviewer very seriously. This message came as a great surprise to me, as it seems to imply that I failed at my tasks and was unfair in my assessment due to some fundamental misunderstandings and misinterpretations on my end. I believe the two main points in the previous message do not address at all the main criticisms raised in my meta-review.
> >
> > ## Experimental section
> > I stand by my claim that
> > > The experimental section could be improved:
> > for the following reasons:
> >    - The experiments section only focuses on sorting (d=1), which is the simplest case of optimal transport.
> >    - No average sorting performance metrics are reported. The experimental results report colomaps of attention patterns on single (relatively small) inputs. It is thus incorrect to consider that these results are (statistically significant) proof of generalisation (out of distribution or not), which contradicts the second bullet point in the previous message:
> > > Figure 3 clearly demonstrates that out-of-distribution generalization occurs in practice
> >
> > ## Clarity and Mistakes in Equations
> > Regarding the third bullet point that states that,
> > > the meta-review inaccurately claims that the response did not address the reviewer’s concerns.
> >
> > I do not see such a claim in my meta review. I just illustrated the lack of clarity and the numerous typos by referring to the fact that some of the typos noted by reviewer mAff are still in the paper after the revision, for instance:
> > > line 275 the iteration index has to be (l+1)
> > > at line 286 some of the zeros must be zero vectors or matrix;
> > I stand by my claim that the paper lacks clarity and still has mistakes after the revision.
> >
> > ## Conclusion
> > As I mentioned in my meta-review,
> > > I believe the idea and the result have high potential, and I encourage the authors to address the abovementioned points in future submissions.
> >
> > Best of luck,
> > your AC

---

> > > ### Public Comment · ~Hadi_Daneshmand1 · 2025-05-02
> > > **Thank you**
> > >
> > > Thank you for confirming that the opening statement of the meta-review is incorrect. A similar mistake appeared in the meta-review I received today, so I wanted to clarify.
> > >
> > >
> > > As someone currently serving as an Area Chair for several conferences, I am aware that ACs have sufficient time to consult with reviewers. AC could just ask the reviewer opinion in AC/reviewer discussions. I assume that if a reviewer raised their score from 3 to 6 with the high confidence score 4, they found the response and revisions convincing and did not identify any major issues.
> > >
> > > The review does not raise any issues related to $d=1$, nor does it mention "average sorting performance" and the review states that a result on even toyish data is interesting. As explained in the response to reviewer mAff,  we can set $|x_i - y_j|^2 = c_{ij}$ in OT, so the number of parameters required to solve the optimal transport problem scales with $n$, not with $d$.

---

### Decision · Program_Chairs · 2025-01-22

Reject